# MEFL: Meta-Equilibrize Federated Learning for Imbalanced Data in IoT

**DOI:** 10.3390/e27060553

**Published:** 2025-05-24

**Authors:** Jialu Tang, Yali Gao, Xiaoyong Li, Jia Jia

**Affiliations:** The Key Laboratory of Trustworthy Distributed Computing and Service, Ministry of Education, Beijing University of Posts and Telecommunications, Beijing 100876, China; auroralu@bupt.edu.cn (J.T.); lixiaoyong@bupt.edu.cn (X.L.); jiajia@bupt.edu.cn (J.J.)

**Keywords:** federated learning, meta-learning, Internet of Things, non-IID data

## Abstract

In the Internet of Things (IoT), data distribution among diverse terminals exhibits substantial statistical heterogeneity. This imbalance can lead to skewness and accuracy degradation, ultimately affecting the generalization ability and robustness of Federated Learning (FL) models. Our work addresses these critical challenges by proposing a novel method, Meta-Equilibrized Federated Learning (MEFL), which integrates meta-learning with gradient-descent preservation and an equilibrated optimization aggregation mechanism based on gradient similarity and variance weighted adjustment. By alleviating the gradient biases caused by multi-step local updates from the source, MEFL effectively resolves the issues of inconsistency between global and local optimization objectives. MEFL optimizes trade-offs between local and global models, and provides an efficient solution for cross-domain data security deployment in IoT scenarios. Comprehensive experiments conducted on real-world datasets demonstrate that MEFL achieves at least 3.26% improvement in final test accuracy, and substantially lowers communication overhead, compared to the existing state-of-the-art baseline methods. The results demonstrate that MEFL exhibits superior performance and generalization capability in addressing personalization challenges with imbalanced non-IID data distributions.

## 1. Introduction

With the rapid development of mobile communication infrastructure and artificial intelligence technology, the data generated at terminal equipment is growing exponentially. As significant advances have been achieved in artificial intelligence (AI) technology, numerous applications have transitioned into intelligence [1]. People are looking forward to obtaining a higher degree of intelligent services with rapid response capabilities [2], working with remote clouds to perform tasks involving local processing and remote coordination. Federated learning (FL) resolves the problem of data isolated islands [3], enabling the design and training of cross-institutional and cross-departmental machine learning models and algorithms. FL enables secure defense mechanisms grounded in privacy protection for cross-domain collaborative training, allowing collaboration among mobile terminals to train a global shared model [4] while keeping data independence.

In the intelligent and interconnected Internet of Things (IoT), the rapid advancement in artificial intelligence computing capabilities has enabled low-complexity, low-cost, and highly efficient computing via sensors, machines, and even crowdsourcing [5]. However, cross-domain collaboration is confronted with significant security challenges, as the stringent data privacy requirements imposed by different entities restrict the modes of data exchange. FL provides a novel training paradigm for cross-domain model collaboration [6], while prioritizing data security. Nevertheless, the extreme imbalance in data distribution poses critical bottleneck issues in deploying FL within real IoT scenarios [7], thereby impacting the performance and efficiency of models in practical applications. It is difficult to establish centralized processing of data in IoT due to the decentralized and diverse nature of the data, the limited connection resources, and the demand for low latency. For example, FL can provide alternative solutions by enabling privacy-preserving intelligence without the need for direct data sharing [8]. However, analyzing certain cases in healthcare applications [9] may involve overrepresented data from specialized hospitals and underrepresented data from other facilities. The imbalance in data distribution across devices [10] may hinder its direct application in guiding diagnosis and treatment.

Traditional FL faces significant challenges due to gradient biases and inconsistent optimization objectives, which are primarily caused by data heterogeneity and device heterogeneity across distributed clients. In the IoT, datasets acquired from discrete terminals are usually non-independently and identically distributed (non-IID) [11], leading to significant skewness in the training model. Gradient biases arise when local gradients computed on divergent data distributions lead to misaligned updates [12], resulting in oscillatory or slow convergence of the global model. Existing work on FL in IoT focuses on generic applications, proposing solutions for application offloading, workload scheduling, and secure communication. They often emphasize the training effect of the global model but overlook fairness among clients, where the global optimal may not align with the local optimal. Inconsistent optimization objectives emerge as clients optimize local loss functions that may conflict with the global objective, potentially causing model divergence or suboptimal performance [13]. These issues collectively degrade the convergence speed, accuracy, and fairness of federated models, particularly in scenarios with highly skewed data distributions or devices with limited resources.

The inherent non-IID nature of client data introduces some fundamental challenges that significantly impair global model performance. In both FedAvg and its personalized variants, clients with large datasets and high update volumes may significantly influence the global model. Such imbalances could potentially hinder convergence and diminish personalization accuracy. Multi-round local training and synchronization conducted under non-IID local data distribution may lead to client drift, which can further result in a decline in performance [14]. The local updates progressively diverge from the global objective, resulting in an aggregated gradient that does not faithfully represent the true descent direction. Thus, standard aggregation methods fail to account for gradient change across clients, leading to suboptimal model updates. Many researchers have begun to focus on personalized federated learning (PFL), which aims to train specialized and customized models [15] for each client, to address the challenges arising from data heterogeneity. By adapting to the distinct data distribution characteristics of individual clients, PFL offers a more flexible learning framework that can better accommodate the requirements of diverse application scenarios.

In realistic IoT scenarios, data distributions across devices are extremely imbalanced and non-IID, leading to skewed local updates, biased gradients, and inconsistent optimization objectives that degrade global model convergence speed, accuracy, and fairness. To achieve secure, efficient, and balanced cross-domain collaboration in the IoT, we address two critical challenges arising from imbalanced non-IID data. (1) Gradient Bias under Multi-Step Local Updates: Local models trained with multiple SGD steps on imbalanced data can introduce higher-order bias terms and slow convergence. (2) Unfair aggregation due to inconsistent optimization objectives: Heterogeneous devices demonstrate divergent update directions and varying performance levels, which causes simple averaging of local models to deviate from the global optimum. We proposed a novel approach that transforms data heterogeneity from a limiting factor into a potential source of advantage. Based on the PFL framework, a global model serves as a starting point, but subsequent training is designed to capture client characteristics and preferences, ultimately delivering a model that better fits the local data distribution. The overall objective is to learn a global model that generalizes well across clients yet can be quickly fine-tuned for individual clients.

**Contributions**: In this work, motivated by the challenges of gradient bias and inconsistent objectives resulting from the imbalanced non-IID data distribution, we propose a novel model, Meta-Equilibrize Federated Learning (MEFL), which extends beyond the current state-of-the-art research. MEFL is a PFL method based on the optimal trade-off between global and local models, ensuring accelerated convergence and enhanced model accuracy across diverse data distributions. This framework is designed with flexibility, enabling the integration of various underlying models according to task requirements. Furthermore, it ensures that the performance of all participating users is not compromised. We summarize our contributions as follows:We propose MEFL based on meta-learning with Gradient Descent Preservation (GDP), which enhances personalization by leveraging client-specific data distribution and neutralizes gradient bias introduced by accumulated local training.We introduce an equilibrized optimizing aggregation algorithm based on gradient variance weighting (MEFL-EOA). Thereby, it effectively balances the impact of each client’s updates by assigning higher weights to clients with more consistent and reliable gradients.Comprehensive experimental results demonstrate that the performance of the proposed method is faster in convergence and achieves higher accuracy than the baselines. MEFL achieves an optimal balance between personalization and generalization capabilities.

The paper is structured as follows. Section 2 reviews and discusses the existing system models in the related domain. Section 3 presents preliminaries and problem formulation. Section 4 introduces the theoretical framework of MEFL and algorithms. Section 5 describes the experimental design. We empirically verify the proposed algorithm and evaluate the results. Finally, we conclude the paper in Section 6, and discuss some implications for future study.

## 2. Related Work

In many horizontal federated learning approaches, a single model is employed for all clients [16]. However, under non-IID settings, this approach leads to convergence problems and degraded prediction performance. The goal of training a single global model on the union of client datasets becomes more challenging with non-IID data distributions, as using one global model struggles to address the statistical challenges posed by such distributions across different clients.

### 2.1. Personalized Federated Learning

The original FL framework proposed by McMahan et al. [17] aims to allow multiple clients to collaborate to train a shared global model without moving local data. Most of the existing FL training approaches are derived from the federated averaging (FedAvg) [17] algorithm, aiming to train a global model that performs well on most FL clients. PFL research is primarily focused on addressing two critical challenges: the issue of poor convergence in highly heterogeneous data and the absence of sufficiently personalized solutions. The PFL approach is highly appealing for intelligent healthcare applications [18], as it enables coordinating multiple clients (e.g., Wise Information Technology of Med) to implement artificial intelligence solutions effectively.

And through multiple rounds of iterative updates, it can obtain a better global model. With the global model for training, clients with scarce local data can gain greater benefits, but clients with sufficient data may not necessarily gain greater benefits. For some tasks, the training effect of the global shared model may not be as good as the local model. Geyer et al. [19] question the utility of a global model that is too far removed from the typical usage of a user. FedRep [20] leverages the distributed computational power across clients to perform multiple local updates on the low-dimensional local parameters for each update of the representation. APFL [21] derives the generalization bound of a mixture of local and global models and finds the optimal mixing parameter.

Research on IoT [22,23,24] focuses on generic applications, where solutions have been proposed for application offloading, workload scheduling, and service migration triggered by user mobility. For example, Konecny et al. [25] propose a stochastic gradient method S2GD, which studies the problem of minimizing the average of a large number of smooth convex loss functions. FedRL [26] builds models of high quality for agents with consideration of privacy under a deep reinforcement learning framework. Yang et al. [27] investigated the convergence rate of federated learning under various scheduling strategies. However, they do not address the relationship among communication, computation, and training accuracy for machine learning applications, which is important for optimizing the performance of machines. However, in personalized FL scenarios where clients possess non-IID data distributions and divergent task requirements, the global objective often conflicts with local objectives.

The PFL-MC framework [28] globally employs multi-center aggregation to dynamically learn multiple global models and locally designs a hierarchical network containing personalized and federated modules to address the issues of data heterogeneity and model heterogeneity. The personalized federated semi-supervised learning system Ferrari [29] is proposed in this study. It generates a client ranking based on the similarity of data distribution through Gaussian KD-Tree, and adaptively determines the model migration strategy and pseudo-label confidence threshold in combination with resource constraints to optimize the quality of pseudo-labels and personalized training. The FedAS framework [30], through federated parameter alignment, integrates local insights into global parameters to enhance their localization, enabling the shared components to learn from historical models to improve local relevance and reduce the impact of parameter inconsistency.

Existing control variate schemes predominantly depend on aggregate statistics rather than gradient alignment. Although variance weighting alone balances contributions by scale, it disregards the direction of the update, potentially resulting in slow or unstable convergence under extreme non-IID heterogeneity. While prior approaches in federated learning personalization, drift correction, and fairness weighting address certain aspects of the challenge, none simultaneously mitigate high-order bias arising from multi-step local updates and evaluate gradient alignment for fair aggregation. This serves as the motivation for our MEFL design.

### 2.2. Mate Learning

In large and complex IoT scenarios, meta-learning [31] offers promising directions for improving the performance and adaptability of machine learning methods. Meta-learning is the process of learning how to learn, which aims to be generalized from a few examples. As one promising direction for meta-learning, initialization-based methods [32] have recently demonstrated effectiveness by learning to fine-tune. The foundational principle of meta-learning algorithms entails identifying and leveraging transferable latent representations extracted from prior tasks, which enhances generalization for unseen tasks and mitigates the problem of data heterogeneity through adaptive learning rates.

Among the various methods, some are focused on learning an optimizer such as the LSTM-based meta-learner [33] and the Meta Networks with an external memory [34]. Meta-learning approaches address few-shot learning, but overtraining can reduce generalizability on new tasks. An entropy-based method [35] with an inequality minimization approach for broader scenarios is proposed to prevent model bias. Existing FL methods suffer from performance drops when clients have few-shot samples in limited data scenarios. A novel framework with two separately updated models and specialized training strategies is proposed to mitigate global data variance and local data insufficiency [36]. However, none of these methods have simultaneously addressed the impacts of data heterogeneity and gradient drift on collaborative model training. Ensemble learning techniques provide more reliable results by combining results obtained by individual models [37].

Another approach aims to learn a good model initialization [38], such that the model has maximal performance on a new task with limited samples after a small number of gradient descents. We adopt a popular and flexible meta-learning approach, which includes well-known algorithms such as MAML [39], Reptile [40] and Meta-SGD [41]. For each task and client in FL, a high-performance model can be learned in only a few gradient steps on new tasks and adapted to a particular task. In the context of PFL, meta-learning based on initialization has shown potential for personalization and dealing with non-IID data. In practice, devices often contribute only sporadically and hold highly unbalanced data. Robust aggregation is thus essential to prevent stragglers or niche clients from skewing the global model [42]. It is essential to consider the computational constraints of IoT devices when implementing solutions for extreme label skew, where certain data classes are underrepresented.

Vettoruzzo A et al. [43] introduced a novel framework that integrates federated learning with meta-learning. By employing a federated modulator to extract contextual information from data batches and generate modulation parameters, the framework dynamically adjusts the activation values of the MAML-based base model, thus enabling model personalization. The pFedLT method [44] introduces pluggable hierarchical transformations based on scaling and shifting operations in the local update stage, and learns these operations through a meta-learning strategy, enabling a single global model to adapt to the data distribution differences among different clients.

While existing meta-learning methods excel in rapid adaptation, they fail to track or correct the bias induced by multiple local SGD steps, which may cause client updates to deviate from the global objective. Consequently, we seamlessly integrate PFL with a meta-learning framework to accelerate convergence while preserving local gradient paths through meta-SGD, thereby neutralizing bias.

## 3. Preliminaries

In light of the heterogeneous characteristics of data in IoT networks, personalization is an essential strategy to improve model performance. This section formulates the preliminary system model and the composition optimization problem, then derives our research contribution in this paper. For the sake of clarity, Table 1 briefly summarizes the meaning of the symbols and notations used throughout this paper.

### 3.1. Personalized FL Model

Following the statistical FL framework and its statistical properties, this work considers N decentralized terminal devices, each possessing private datasets restricted from accessing the others, donated by $D_{i}={\left\{(x_{i},y_{i})|X\times Y\to R\right\}}_{i=1}^{N}$, where $X\in R^{d}$ is the input domain and Y is the corresponding label domain. For client i∈N, we define the model parameters as the vector $w\in R^{d}$, and train the models collaboratively by exchanging learned parameter vectors $w_{i}$ instead of the original data, and $l_{i}\left(\cdot \right)$ is the loss function with a randomly selected feature sample. The true risk at local distribution $D_{i}$ quantifies the extent of error between predicted and actual outcomes, which is indicated by empirical risk $L_{i}\left(w_{i};D_{i}\right)=E_{\left(x,y\right)∼D_{i}}l_{i}\left[w;(x,y)\right]$. Each client utilizes private data to learn and update the local model, which is derived from the global model. Then, the server performs the asynchronous aggregation and proximal optimization among randomly selected clients N⊆N. The updates are as follows: (1)$$\begin{array}{cc} Local:\; & w_{i}^{t+1}=w_{g}^{t}-\eta g_{i}^{t}, \end{array}$$(2)$$\begin{array}{cc} Global:\; & w_{g}^{t+1}=\sum_{k=1}^{N}p_{k}w_{k}^{t+1}, \end{array}$$(3)$$\begin{array}{cc} \; & g_{i}^{t}=\nabla_{w_{i}}L_{i}\left(w_{i}^{t};D_{i}\right)=\frac{1}{\left|D_{i}\right|}\sum_{\left(x,y\right)\in D_{i}}\nabla_{w_{i}}l_{i}\left(w_{i};x,y\right),\\ s.t.\; & p_{k}=\frac{\left|D_{k}\right|}{\left|D\right|},\;\left|D\right|=\sum_{k=1}^{N}\left|D_{k}\right|, \end{array}$$
where η is the learning rate, and $g_{i}^{t}$ is the gradient computed at the *i*th (i∈N) device. In Equation (Equation 2), $p_{k}$ is the empirical distribution of the data samples, where $\left|D_{i}\right|$ is the amount of dataset $D_{i}$, and $\left|D\right|$ is the number of total data samples. All clients independently perform multiple iterations of local updates before uploading their local models, which accelerates the convergence rate.

### 3.2. Problem Formulation

In conventional FL with algorithms like FedAvg [17], clients collaboratively train a single global model by minimizing a unified empirical risk objective. The loss function is commonly utilized for performance evaluation, and the objective is to find the optimal parameter by minimizing the following empirical risk overall clients: (4)$$\min_{w}L_{g}\left(w;D\right)=\sum_{i=1}^{N}p_{i}L_{i}\left(w;D_{i}\right),$$
where $p_{i}$ represents the relative contribution of client *i*, and each client optimizes its local model using gradient descent on its local dataset. However, this results in model drift, as each client minimizes $L_{i}\left(w;D_{i}\right)$ instead of the true global objective $L_{g}\left(w;D\right)$. The global model optimized through averaging local updates fails to align with the true optimization directions of individual clients. The objective mismatch causes an aggregation bias, where the global update may not correctly approximate the optimal descent direction. And optimizing local models on non-IID data causes local optima to diverge from the globally optimal solution. Therefore, we aim to minimize the discrepancy between local gradients and the true global gradient to ensure stable convergence.(5)$$\min_{w}\sum_{i=1}^{N}\left|\nabla L_{i}\left(w;D_{i}\right)-\nabla L_{g}\left(w;D\right)\right|.$$

### 3.3. Gradient Bias

FL algorithms commonly use multiple local training steps before global aggregation, causing the aggregated global model to deviate from the optimal direction for the overall objective. Each client starts with the same global model $w_{g}$ but then performs multiple local updates on their local data to adjust the model. When client data distributions are non-IID, these local updates $g_{i}=w_{i}^{t+1}-w_{i}^{t},\left(t\geq 0\right)$ can lead to gradients biased towards the local data distribution. Consequently, the gradients computed at these locally adapted models no longer accurately represent the gradient at the global model’s parameter setting. We consider that the local update incorporates higher-order terms from the Taylor expansion of $L_{i}$ around $w_{i}$, for instance: (6)$$\begin{array}{cc} w_{i}^{t} & =w_{i}^{t-1}-\eta \nabla L_{i}\left(w_{i}^{t-1};D_{i}\right), \end{array}$$(7)$$\begin{array}{cc} w_{i}^{t+1} & =w_{i}^{t}-\eta \nabla L_{i}\left(w_{i}^{t};D_{i}\right), \end{array}$$(8)$$\begin{array}{cc} \nabla L_{i}\left(w_{i}^{t};D_{i}\right) & =\nabla L_{i}\left(w_{i};D_{i}\right)+\nabla^{2}L_{i}\left(w_{i};D_{i}\right)\left(w_{i}^{t}-w_{i}\right)+O\left({∥w_{i}^{t}-w_{i}∥}^{2}\right), \end{array}$$
where $t\geq 1,w_{i}^{0}=w_{g}$. Therefore, the effective update in the first two steps is yielded as: (9)$$\begin{array}{cc} w_{i}^{\left(2\right)} & \approx w_{i}-2\eta \nabla L_{i}\left(w_{i};D_{i}\right)+\eta^{2}\nabla^{2}L_{i}\left(w_{i};D_{i}\right)\nabla L_{i}\left(w_{i};D_{i}\right), \end{array}$$(10)$$\begin{array}{cc} g_{i} & =w_{i}^{\left(2\right)}-w_{i}\approx -2\eta \nabla L_{i}\left(w\right)+\eta^{2}\nabla^{2}L_{i}\left(w\right)\nabla L_{i}\left(w\right), \end{array}$$
where the extra term $\eta^{2}\nabla^{2}L_{i}\left(w\right)\nabla L_{i}\left(w\right)$ does not appear in a standard single-step gradient at $w_{i}$. It shifts the client’s effective update away from $-2\eta \nabla L_{i}\left(w\right)$, introducing bias. Due to the nonlinearity of the loss function and the fact that clients update their models using gradients computed at intermediate parameters, it introduces bias in the gradient estimation due to the discrepancy. This bias can accumulate over multiple local updates, leading to divergence from the optimal global model and affecting the convergence rate.

Local updates computed on heterogeneous data inherently introduce biased gradients, which propagate to the global model during aggregation. As shown in Figure 1, this misalignment introduces a gradient bias that accumulates across communication rounds, exacerbating convergence instability and suboptimal generalization. To address these challenges, we formally define the problem and establish optimization objectives that align local and global training dynamics. Each client updates its model towards minimizing its own local objective, which does not necessarily align with the global objective. Therefore, the optimization objective in Equation (Equation 4) is instantiated as follows: (11)$$\min_{w,\alpha }\left\{E_{\left(x,y\right)∼D_{i}}\left[L_{i}\left(w^{\prime }\right)\right]:=\frac{1}{\left|D_{i}\right|}\sum_{i=1}^{N}L_{i}\left[w-\alpha \circ \nabla L_{i}\left(w\right)\right]\right\},$$
where α is a vector of the same dimension as *w*, and a learnable parameter adaptation rate that enables fast personalization. Learn *w* to serve as a robust initialization for fast adaptation to any client. For client *i*, learn $w_{i}$ by fine-tuning *w* on its private dataset $D_{i}$, minimizing task-specific loss $L_{i}\left(w_{i};D_{i}\right)$. This enables rapid client adaptation by learning both the model initialization and adaptive per-parameter learning rates.

## 4. Method

This section proposes a PFL framework based on meta-learning and analyzes its statistical properties, motivated by the challenges of inconsistent optimization objectives and gradient biases. We conduct a theoretical analysis demonstrating that this architecture exhibits robust convergence and personalization capabilities in non-IID environments.

### 4.1. Architecture of MEFL

To enable the personalization and generalization capabilities of a personalized model to mutually facilitate each other, we propose a PFL model that integrates meta-learning and a variance weighting mechanism based on gradient similarity. The model is designed based on the fundamental knowledge outlined in Section 3, which is illustrated in Figure 2. To better cope with heterogeneity, it is common to consider learning device-specific personalized models that enable rapid adaptation to novel tasks. Consequently, the Meta-Equilibrize Federated Learning (MEFL) model is constructed on a neural network architecture with an initial global model, which handles non-convex problems well. Instead of forcing all clients into a single global model, we integrate Meta-SGD-based adaptation to personalize client models, improving local performance while retaining a well-generalized global representation. This approach is beneficial in non-IID settings with limited data.

### 4.2. Meta-Learning with Gradient Descent Preservation

By leveraging the capabilities of the model-agnostic meta-learning algorithm Meta-SGD [41], the global model in federated learning not only quickly adapts to the local data of the clients but also alleviates overfitting. Each task provides a support set $D^{s}$ and a query set $D^{q}$, both containing labeled data. Notably, the support and query sets are disjoint, ensuring an unbiased assessment of the learned model’s performance. It formalizes as a bi-level optimization problem with the following components: a meta-initialization parameterized by *w* and a few gradient updates performed with adaptive learning rate α; meta-optimization leveraging client-specific parameters $w_{i}\left(i=1,\ldots ,N\right)$ derived from *w* local adaptation. The details are shown in Algorithm 1. In the inner update, it trains the model on the support set and yields parameters $w_{i}$, retaining the computational graph. The resulting model is subsequently evaluated on the query set, producing a test loss $L_{i}^{q}\left(w_{i};D_{i}^{q}\right)$ in the outer loop, then computes the meta-gradient and backpropagates to the initial parameters. Thus, we replace Equation (Equation 1). For multiple local update steps τ≤E, this update is iterated as: (12)$$w_{i}^{\tau }=w_{i}^{\tau -1}-\alpha \circ \nabla L_{s,i}\left(w_{i}^{\tau -1};D_{i}^{s}\right),$$
where $\alpha \in R^{d}$ is an adaptive vector of learning rates and ∘ denotes element-wise multiplication. We propose the approach of Gradient Descent Preservation (GDP), which counteracts the gradient bias resulting from extended local training. The local gradient updates on the client side are consistently anchored to the global initial parameters $w_{g}$, effectively mitigating the path dependency bias introduced by multi-step local updates. Symbolic updates retain dynamic information, ensuring that subsequent gradient computations are grounded in the initial gradient. The updated trajectory of the inner loop is implicitly embedded within the computational graph, enabling the meta-gradient of the outer loop to effectively reflect the contribution of personalized adaptation to the global model.
**Algorithm 1:** Meta-GDP: Personalized update algorithms based on meta-learning with gradient descent preservation.
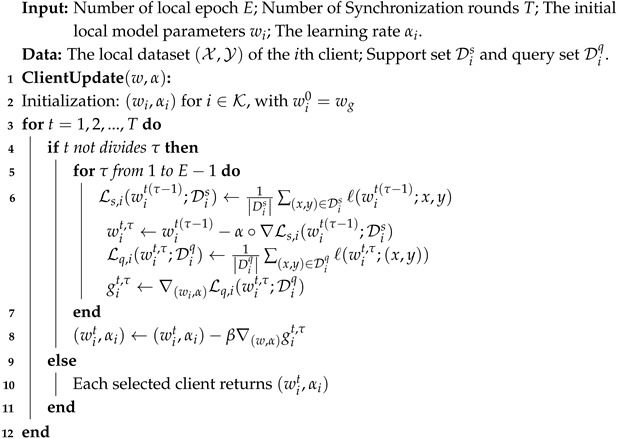


After completing *E* − 1 local update epochs, we obtain the updated local model $w_{i}^{t}$ on client *i* in round *t*. Then, we evaluate it on the cross-client datasets and compute the gradient concerning the shared initial model parameters by leveraging the computational history recorded during the aforementioned GDP process.(13)$$g_{i}^{t,\tau }\leftarrow \nabla_{\left(w_{i},\alpha \right)}L_{q,i}\left(w_{i}^{t,\tau };D_{i}^{q}\right).$$
After τ local updates, the server collaboratively aggregates the global model $w_{g}^{t}$ from the local models $w_{i}^{t}$ on those randomly selected clients. This model strengthens personalization by taking advantage of client-specific data distributions and counteracts the gradient bias resulting from extended local training.

### 4.3. Equilibrized Optimizing Aggregation

We introduce an Equilibrated Optimization Aggregation (EOA) algorithm combining gradient similarity and variance-based weighting to ensure stable, fair, and directionally aligned global updates. (See Algorithm 2). It means that clients with more consistent and reliable gradients have a proportionally greater impact on the global model. This dual mechanism improves convergence speed and robustness. Each client receives the global model $w_{g}$ and fine-tunes it on its local data using Meta-GDP. The client updates its model to obtain $w_{i}$ after a set number of epochs *E* and computes the unbiased gradient $g_{i}=w_{i}-w_{g}$. We define the average of the gradients obtained from previous rounds as the estimated global gradient G. To further stabilize the estimate, the global gradient direction is updated using the exponential moving average (EMA). For each client, the similarity score reflecting directional alignment is computed with a minimum value ϵ as: (14)$$\begin{array}{cc} G & =\gamma \cdot G+\left(1-\gamma \right)\cdot \frac{1}{N}\sum_{i\in N}g_{i}, \end{array}$$(15)$$\begin{array}{cc} S_{i} & =\frac{g_{i}\cdot G}{∥g_{i}∥\cdot ∥G∥+\epsilon }. \end{array}$$
At each round *t*, client *i* computes a local gradient $g_{i}^{t}$ and variance of previous gradients $\sigma_{i}$, which can be performed incrementally using an online update formula. We assume that the variance of the client’s gradient is negatively correlated with its similarity $S_{i}$ to the global gradient (i.e., the higher the similarity, the smaller the variance).(16)$$w_{i}\propto \frac{1}{\sigma_{i}^{2}}\propto S_{i}.$$
For each client *i*, estimate the variance $\sigma_{i}^{2}$ of its gradients over recent training rounds as $\sigma_{i}^{2}=\frac{1}{T}\sum_{t=1}^{T}{∥g_{i}^{t}-{\bar{g}}_{i}∥}^{2}$. Instead of using the raw variance, we update the variance with an exponential moving average: (17)$$\sigma_{i}\leftarrow \lambda \sigma_{i}+\left(1-\lambda \right)\sigma^{2},$$
where λ is a smoothing factor controlling how much past gradients influence the variance. The raw optimal weights $\rho_{i}$ are calculated based on both gradient similarity and variance: (18)$${\tilde{\rho }}_{i}=\frac{S_{i}}{\sigma_{i}+\epsilon },$$
where ϵ is a small constant to ensure numerical stability. Then, we evaluate it with a data batch to compute the gradient $g_{i}^{t}$ with respect to $w_{i}^{0}$. Normalize the weights so that they sum to one across all participating clients: $\rho_{i}=\frac{{\tilde{\rho }}_{i}}{\sum_{j=1}^{N}{\tilde{\rho }}_{j}}$.
**Algorithm 2:** MEFL-EOA: Meta-Equilibrated Federated Learning with Equilibrated Optimization Aggregation.
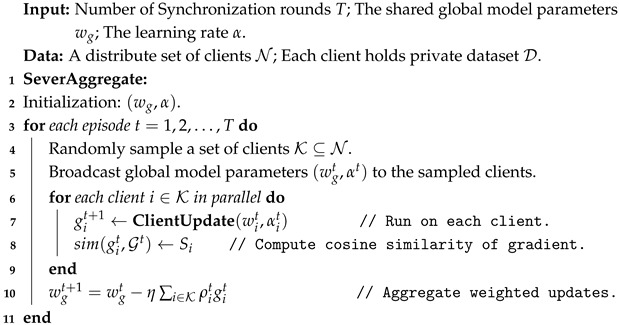


Clients with consistent directions and stable updates will naturally be assigned higher aggregation weights, thereby enhancing the convergence efficiency and quality of the global model. A global EMA gradient vector representing the smooth synthesis of the update directions of each client in history serves as the global direction benchmark. Cosine similarity $S_{i}$ only cares about the directional relationship between two vectors and is not disturbed by the length magnitude of the vectors. When $S_{i}\approx 1$, it indicates that the update direction of the client is highly consistent with the global direction, suggesting that the data distribution of this client is well matched with the overall trend. And $S_{i}\approx 0$ reflects that the two clients are almost orthogonal, and the update has no obvious connection with the global objective. The negative value means client update is contrary to the global direction. In this way, MEFL can automatically amplify the contributions of those beneficial for global update clients, suppress updates with high noise and low alignment, thereby enhancing the convergence speed and personalization quality of the global model.

The server weighted the aggregates of the updates to produce a new global model to replace Equation (Equation 2), which is agnostic to the underlying architecture.(19)$$w_{g}^{t+1}=w_{g}^{t}-\eta \sum_{i\in N}\rho_{i}g_{i}^{t},$$
where η is the global learning rate. This integration ensures that the aggregation process remains unbiased and that client updates contributing stably have a higher influence on the global update. While adding gradient similarity computations increases complexity slightly, it can be cost-effective compared to the risk of aggregating misaligned, noisy updates. An optimal desn igbalances these factors without overburdening client or server resources.

## 5. Results

This section presents a concise and precise account of the experimental results, their interpretation, and the conclusions that can be derived from the experiments.

### 5.1. Experimental Setup

To evaluate the performance of MEFL across different tasks and models in IoT networks, we present numerical results obtained from using simulated real-world non-IID datasets. These results demonstrate that MEFL achieves faster convergence, higher accuracy, and lower system overhead compared to traditional federated learning approaches. We adopt a rigorous data partitioning strategy to ensure that the training, validation, and testing processes of the model can comprehensively evaluate the performance of our proposed method. We assess MEFL in a more realistic scenario to verify its feasibility and robustness in practical applications. We randomly selected 80% of all available clients to form the training set, of which 10% was allocated as the validation set for tuning hyperparameters and mitigating overfitting, and the remaining 10% was reserved as the testing set to evaluate model generalization.

**Datasets.** Data from RedIoT devices generally exhibit extreme heterogeneity and a long-tail distribution characterized by an imbalanced representation. Therefore, we select the following datasets and design the corresponding non-IID settings: Fashion-MNIST [45], CIFAR-10 [46], and Sentiment140 [47]. The Fashion-MNIST dataset realizes an imbalanced distribution of real-world features by aggregating handwritten characters/digits from distinct authors based on the MNIST dataset [48]. A natural skew in feature distribution across participants arises due to the association of each participant with distinct authors. The CIFAR-10 dataset is a relatively comprehensive object image dataset that contains 60,000 color images of 32×32 pixels. Sentiment140 consists of 1.6 million Twitter messages with emoticons, which are used as noisy labels for sentiment classification.

To simulate label imbalance, we map classes of images to different device types to simulate sensor type differences and perform Non-IID data partitioning for each client using Dirichlet distribution [49] by sampling $p_{k}∼Dir\left(\delta \right)$ over labels and allocating a $p_{k,i}$ proportion of the instances of class *k* to party *i*. And we sample q∼Dir(δ) and allocate a $q_{i}$ proportion of the total data samples to $P_{i}$. Each client contains at least 2–3 classes to avoid a completely single-class distribution. For each client, the local data are divided into the support set and query set. Additionally, we systematically varied the proportion of data in the support set, investigating the efficiency of MEFL adaptation for new users with limited data.

**Baseline.** We perform a comparative analysis against the classic global FL, various PFL, and FL methods optimized for data heterogeneity. Our baselines include standard baseline FedAvg [17], which naively averages client updates after multiple local SGD steps; FedProx [50], which partially mitigates drift incorporating a regularization proximal term; perFedAvg [51], which extends client-specific fine-tuning of a global model initialization; and pfedMe [52], which uses a regularization with a Moreau envelope function. To validate the theoretical findings, we report the optimization and training performance, as well as the generalization performance on local data.

**Evaluation Metrics.** We evaluate the proposed methods with various network architectures in non-IID FL settings. We evaluate the test accuracy of algorithms on test datasets and observe the dynamics of training loss to compare convergence speeds. To verify the impact of multiple iterations of the local model and considering the limited computation capacity on edge devices, we set different local epochs for each method. By default, we set the parameters as B=64, E=10.

Based on the metrics introduced by Divi et al. [53], we assess the personalized federated learning model from the perspective of client-side performance enhancement. To quantify the accuracy improvement for each user attributable to personalization, we compute the Quantum of Improvement (QoI) metric as follows:(20)$$F_{i}=PA_{i}-\max\left(GA_{i},LA_{i}\right),$$
where $PA_{i}$ represents the accuracy of the personalized model, $GA_{i}$ denotes the accuracy of the global model on client i, and $LA_{i}$ refers to the accuracy of the local training model. For clarity in presentation, all instances of $F_{i}$ in the equations are denoted as QoI. Due to variations in data distribution quality across clients, the FL global model may lead to a reduction in the accuracy of the personalized model for certain users, potentially resulting in negative values of QoI. This indicates that personalization does not guarantee performance improvements for all users’ local models.

Additionally, in the evaluation of both classical FL methods, a systematic approach is adopted to ensure consistency and fairness in performance assessment. For methods focusing on generating global models, the performance of the global model is directly utilized for assessing the accuracy of personalized models. In contrast, for approaches that generate explicit personalized models, the simple average aggregation of personalized models is employed to evaluate personalization capabilities. This evaluation strategy highlights the importance of considering both global and personalized aspects in federated learning research. Therefore, we can gain deeper insights into the trade-offs between centralized performance and client-specific adaptability.

**Implementation Details.** We conducted experiments on the Fashion-MNIST dataset using a CNN model consisting of two 5 × 5 convolutional layers with 32 and 64 channels. After each convolutional layer, a ReLU activation function and a 3 × 3 max pooling layer with a stride size of 2 are connected in sequence. The model includes two fully connected layers applying a ReLU activation function and random dropout regularization. We implemented a lightweight version of ResNet-18 for the CIFAR-10 dataset by removing the last two residual blocks from the original architecture. This modification significantly reduced the number of parameters, thereby addressing the constraints imposed by federated communication. Additionally, we substituted the final layer with a fully connected layer tailored for 10-class classification. The model we developed for the Sentiment140 dataset utilizes a two-layer Multilayer Perceptron (MLP) architecture with a specific configuration 512→256→1 and incorporates the ReLU activation function. For feature extraction, the model employs Term Frequency-Inverse Document Frequency (TF-IDF) representation, which is further processed using Principal Component Analysis (PCA) to reduce the dimensionality to 512 while preserving 95% of the variance information.

**Hyperparameters.** We employed grid search to determine the optimal meta-learning hyperparameters (α,β) for various datasets. The inner learning rate α governs the extent of local fast adaptation, while the outer learning rate β controls the intensity of global model updates. A higher value of α may amplify discrepancies in local updates, theraby increasing the communication rounds and resulting in fluctuations in the global accuracy curve. The outer learning rate β determines the initial parameters for all clients in the subsequent round. The global learning rate η not only influences the speed of local fitting in each round but also affects the number of convergence rounds by introducing gradient bias and altering the noise level in global aggregation. For each dataset, we control the convergence speed and stability of FL by fine-tuning the active local batch size, the number of training rounds, and the client sampling ratio per round. Furthermore, we define the historical smoothing weight λ for computing the global EMA gradient and the EMA decay factor for estimating client gradient variance. An excessively low γ may render the EMA overly sensitive, leading to intensified convergence oscillations. Conversely, an excessively high γ may result in a delayed response to new gradient information, thereby diminishing the speed of personalization. To ensure a fair comparison, all the examined algorithms were run for an identical number of iterations. The number of iterations was set to 500 by default unless otherwise specified. Additionally, the meta-learning rate α∈ [0.001, 0.01] is dynamically tuned using grid search. The global learning rate η∈ [0.001, 1] is adjusted according to the amplitude of training loss.

### 5.2. Evaluation Results

We incorporate our MEFL method into the latest FL methods and perform comprehensive comparative experiments to evaluate improvements in accuracy. Specifically, we implement computation graphs to preserve gradient descent for $g_{i}^{t,\tau }$ by Equation (Equation 13) during the automatic differentiation process. We demonstrate effectiveness in enhancing model accuracy and personalization capabilities through a detailed performance analysis before and after integrating the GDP and EOA methods. Primary validation relies on the performance of the final model assessed on the server side under real IoT conditions. Furthermore, we systematically design multiple levels of data heterogeneity to rigorously evaluate the robustness of the model and its broad applicability in diverse scenarios.

**Performance Comparison.** We compare the performance of the MEFL with four baseline frameworks on several non-IID datasets. We aim to solve heterogeneous problems caused by unbalanced data amounts and label distribution. Due to the characteristics of discrete data distribution and inconsistent optimization objectives, the global model suffers from poor aggregation performance and slower convergence speed. In various non-IID settings, there is a gap between the accuracy of existing methods. Label-skewed distributions present the greatest challenge, and the case of quantity-skewed distributions has minimal impact.

For empirical evaluation, we consider CIFAR-10 and Sentiment140 with different heterogeneous distributions. Specifically, CIFAR-10 is set under label-imbalanced conditions (as $p_{k}∼Dir\left(0.5\right)$). Sentiment140 exhibits both quantity and label imbalances (following q∼Dir(0.5), $p_{k}∼Dir\left(0.5\right)$). MEFL converges both faster and to a higher final accuracy than the baselines in all cases of non-IID settings. As depicted in Figure 3, MEFL demonstrates a substantial superiority over other baseline models with respect to convergence speed and final accuracy. We observe that focusing on personalized capabilities (pfedMe, perFedAvg, and MEFL) can improve the performance of the global model. MEFL increases the final accuracies by 7.26–15.18% compared to FedAvg, accounting for the effects of inconsistent optimization directions.

MEFL exhibits significantly accelerated convergence, requiring over 50% fewer communication rounds to achieve 80% accuracy in comparison to FedAvg. FedProx and FedAvg exhibit slower convergence rates owing to inadequate local training. The comparison of training loss over communication rounds is presented in Figure 3c,d. MEFL drives the training loss down more rapidly and achieves lower asymptotic curves than all baselines across heterogeneous dataset splits. It demonstrates that MEFL mitigates client drift and aggregates more informative updates under data heterogeneity. All models demonstrate consistently higher stability and accuracy on CIFAR-10. This may be explained by the highly non-IID nature of the data distributions in the Sentiment140 dataset, as well as the insufficient representativeness of the sampled shared dataset.

The impact of varying local update steps on convergence is illustrated through the convergence curve by adjusting the local training batch size *B* and the number of local training epochs *E*. Fashion-MNIST is configured with significant quantity imbalance (following q∼Dir(0.5)). As shown in Figure 4, the final outcomes with E=5 perform better than E=20. The performance reduction may result from the increasing gradient biases with more local updates, increasing the divergence between local and global models. However, MEFL outperforms all baselines in both cases and exhibits greater stability with a smaller fluctuation range of 0.02–1.36%. Moreover, even slight variations in the optimization process across rounds may negatively affect the final performance. Meanwhile, we compare the convergence curves (Figure 3a,b and Figure 4a) across several benchmarks to demonstrate the accuracy and convergence speed under diverse data distributions. We can see that the imbalanced label distribution most influences the accuracy of the FL methods. Comparing the overall performance, MEFL outperforms the baselines significantly, as it restrains bias through gradient revision and similarity weighting.

**Ablation Experiments.** We illustrate the improvements of gradient descent preservation and equilibrated similarity aggregation algorithms in terms of the test accuracy over sufficient communication rounds in Table 2. The method only deploying the Meta-GDP algorithm is denoted as w/o EOA, while the one leaving out meta-learning and considering similarity-weighted aggregation is denoted by w/o GDP. As the support fraction *P* increases, the accuracies of both Meta-GDP and EOA improve in nearly all scenarios. Nevertheless, the rate of improvement for both Meta-GDP and EOA exceeds that of MEFL. For instance, on the CIFAR-10 dataset, when the *P* rises from 20% to 90%, the accuracy improvement of MEFL is 0.43%, lower than the respective improvements of w/o EOA (0.94%) and w/o GDP (2.11%). This indicates that the meta-learning algorithm exhibits superior generalization capabilities and can effectively adapt to new clients with sparse data resources. The ablation study results of different datasets, which consider the performances of Meta-GDP and MEFL-EOA separately, demonstrate that both approaches show improvements.

The quantitative analysis in the article regarding the local epoch and batch size on stability and divergence mainly stays at the level of mean accuracy and standard deviation. As shown in Table 3, the accuracy of MEFL only drops minimally, from E=5 to E=20, among all methods. The standard deviation of MEFL was the smallest among all settings 0.15∼0.32%, indicating that it maintained extremely high consistency under different local training intensities. The mean accuracy rate of MEFL consistently leads, reaching a peak of 98.21 ± 0.15% at E=10 and B=64, significantly outperforming other methods with the least fluctuation. The smaller and consistent drops indicate that this method can maintain high performance under different local training intensities, suggesting that the personalization process does not introduce additional divergence risks.

The QoI curves provide a quantifiable illustration of how effectively each client is personalized, encompassing both performance gains and the nature of model deviations required to achieve such improvements. Owing to the potential presence of negative values, the direct utilization of the evaluation metric could lead to misinterpretation of the results. Consequently, in our assessment, we restrict our consideration to clients exhibiting positive absolute QoI values on a provisional basis. To further quantitatively assess the personalized benefits of the PFL model, we present the density distribution of client QoI on the Fashion-MNIST dataset. This kernel density estimate illustrates the distribution of per-client personalization gains. As shown in Figure 5a, MEFL not only maximizes average personalization gain, but also ensures that gain is uniformly achieved across almost all clients, underscoring its superiority in personalized federated learning. MEFL’s curve is shifted farthest to the right, indicating the highest average QoI. w/o GDP or w/o EOA shifts the curve leftward and slightly flattens it, confirming that both components contribute to elevated and more uniform personalization gains. pFedMe also improves over perFedAvg but does not match the full MEFL’s level or consistency. This ordering reflects each method’s increasing capacity to personalize. The sharper, taller peak of MEFL shows that most clients receive consistently high QoI, with low variance. In contrast, perFedAvg’s flatter, broader shape signals more variability and less reliable personalization across clients. All of those curves validate the superior performance of MEFL compared to other methods in enhancing both local and global model performance.

**Fairness and Overhead.** In MEFL, the computation complexity of three additional mechanisms, including GDP, EMA and gradient similarity, is separately analyzed as *O*(*d*), *O*(*d*), and *O*(*Ed*). The total complexity of a single round of MEFL locally can thus be expressed as *O*(2*Ed*+2*d*). Theoretically, the local computational overhead resulting from the additional mechanisms introduced by MEFL increases 20%. However, owing to the substantial acceleration of model convergence at the aggregation stage achieved by MEFL with a significant reduction in the number of communication rounds, a net saving in computing resources can still be realized across the entire training process. We define the system budget as the aggregate FLOPS across all devices and the cumulative number of bytes transmitted to and from the server. As shown in Figure 5b, we evaluated the system overhead required per round to achieve 90% accuracy on the target test set using various methods. Regarding computational cost, MEFL demonstrated the lowest consumption of communication resources among all approaches. Computational cost is expressed in FLOPs and communication size in normalized bytes. The incremental computational cost associated with MEFL for achieving enhanced model performance remains within an acceptable range. MEFL reduces per-round communication over 75%, which arises from accelerated convergence. The combined system overhead of MEFL is the lowest among all methods when normalized by rounds to convergence. Overall, these results confirm that MEFL’s personalization gains do not come at the expense of prohibitive system overhead.

MEFL reaches 90% of its own final loss in fewer communication rounds. And the standard deviation of per-round accuracy is smallest for MEFL, indicating that similarity-weighted aggregation successfully dampens the fluctuations induced by data heterogeneity. These results validate that our method MEFL yields significant gains in both convergence speed and final model quality across diverse non-IID scenarios. In addition, we summarize the number of communication rounds required to reach specific accuracy milestones on Fashion-MNIST under a heterogeneous label setting, as shown in Table 4, as well as the final personalized convergence accuracy for all methods, with settings E=10, B=64 and P=80% for all methods. To reach an accuracy of 90%, MEFL requires only 14.68∼36.76% of the communication rounds compared to the respective baselines, which indicates a significant reduction in the communication costs. Meanwhile, MEFL achieves a final convergence accuracy of 98.03%, outperforming other methods. MEFL achieves a relative loss reduction of 28∼53% at convergence compared to the baselines, indicating faster and more stable training.

For datasets with non-IID settings, a well-chosen global step size ensures rapid incorporation of client differences without inducing excessive oscillation. In practice, balancing the local and global step sizes has proven effective, achieving a favorable trade-off between personalization and global convergence stability. We performed a grid search for the learning rate and determined that the optimal balance setting of (0.001, 0.0001) achieves the best performance. This configuration enables the model to adapt to distribution drift more efficiently while minimizing the introduction of additional noise, thereby ensuring rapid and smooth convergence of the global update curve. Additionally, the EMA smoothing factor γ plays a more significant role in influencing the stability of the global model. For high γ, EMA exhibits strong memory retention for old gradients, resulting in a slower response to abrupt changes in new gradients. Consequently, $S_{i}$ (similarity score weight) becomes more stable, which may cause overly conservative personalized updates and lead to a marginal reduction in QoI gain. Conversely, for low γ, EMA demonstrates higher sensitivity, causing $S_{i}$ to fluctuate significantly due to local noise. While this increases the personalization opportunities for some clients (enhancing QoI), it compromises overall stability.

We have generated the accuracy data for each round of 300 experiments on the Fashion-MNIST dataset under different local learning rates and EMA smoothing factors γ. To optimize performance, we conducted a grid search within the range $\eta \in \left[0.001,1\right],\gamma \in \left[0.5,1\right]$ and selected the optimal value based on both final global accuracy and the number of convergence rounds. As shown in Figure 6, a smaller value of (γ=0.7) allows for a faster response to the gradients from each client in the subsequent round. It may also cause the EMA to become excessively sensitive, thereby exacerbating convergence oscillations. Conversely, an excessively high value leads to a delayed response to new gradient information, which in turn reduces the speed of personalization. The parameter determines the step size of each local SGD update. A larger step size accelerates local fitting within each round but also amplifies gradient shifts, thereby increasing global aggregation noise and extending the number of convergence rounds. In contrast, a smaller step size facilitates smoother global updates but necessitates additional rounds to achieve comparable improvements in global accuracy.

## 6. Conclusions

In this work, we address the issues of skewness, accuracy degradation, and insufficient local personalization caused by non-IID data. Multiple local steps and insufficient client sampling both slow convergence and degrade accuracy under non-IID splits. We extend the PFL frameworks with meta-learning and equalized optimization aggregation method. The stability of the novel method MEFL has important implications for real-world IoT under extreme data imbalance, beyond raw accuracy and speed. In practice, devices often contribute only sporadically and hold highly unbalanced data; robust aggregation is thus essential to prevent stragglers from skewing the global model. MEFL excludes anomalous updates and smooths the global optimization landscape, thereby ensuring that representative patterns with broad coverage dominate the global objective to promote fairness.

While MEFL advances the state of PFL, it introduces two computational overheads: (1) the need to record and back-propagate through local update histories (GDP) and (2) the similarity computation for gradient vectors during aggregation. Future research should explore lightweight approximations, for instance, using low-dimensional gradient sketches or selectively applying full preservation only on clients exhibiting large drift. In future, integrating DP noise into preserved gradients and weights with cryptographic secure-sum protocols could yield privacy-guaranteed MEFL variants. Rather than selecting clients uniformly at random, future methods could proactively choose clients whose gradient similarity profiles indicate the greatest potential to reduce global loss. By addressing these challenges, MEFL can evolve into a broadly applicable and efficient framework.

## Figures and Tables

**Figure 1 entropy-27-00553-f001:**
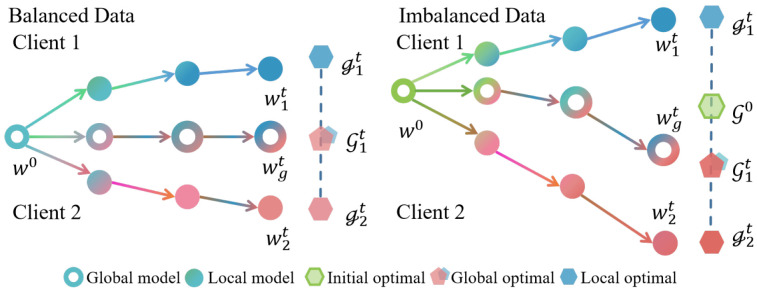
An imbalanced data distribution leads the model aggregation to become skewed, which results in the global model updates deviating from the initial optimization objective.

**Figure 2 entropy-27-00553-f002:**
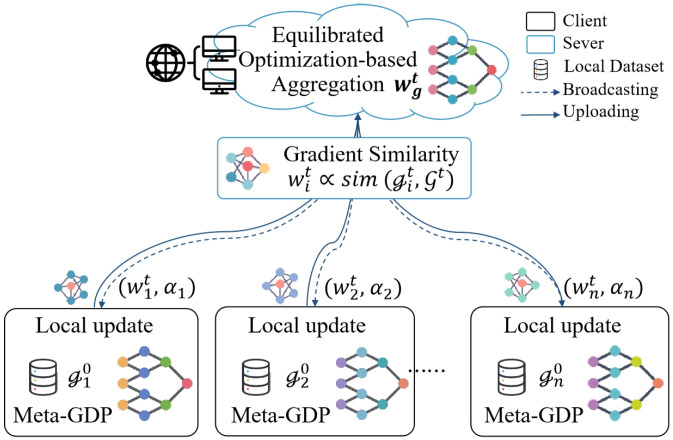
The architecture of MEFL incorporates local training with Meta-GDP and model aggregation with EOA. Each client trains private model with local dataset. To alleviate the adverse effects of accumulated gradient skew, the client leverages a gradient preservation algorithm to improve the meta-learning update process. The importance of parameters is quantified with gradient similarity, and the global optimization objective is unified through weighted aggregation. A typical workflow contains model distributing, local training, model updating, and model aggregating.

**Figure 3 entropy-27-00553-f003:**
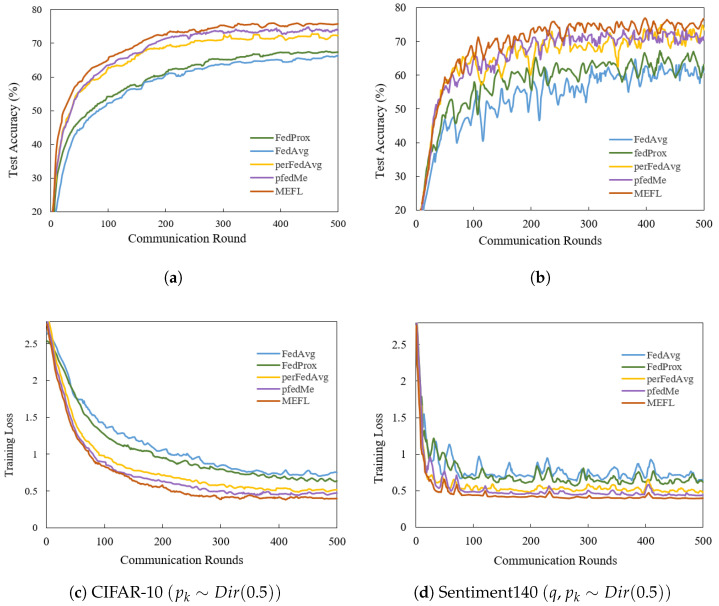
Test accuracy and training loss curves over communication rounds under non-IID datasets (with quantity and label imbalance) for MEFL compared with FedAvg, FedProx, perFedAvg, and pfedMe. (**a**,**c**) Convergence curves of test accuracy and training loss on CIFAR-10. (**b**,**d**) Convergence curves of test accuracy and training loss on Sentiment140. Comparing the overall performance, MEFL exhibits faster convergence and higher accuracy.

**Figure 4 entropy-27-00553-f004:**
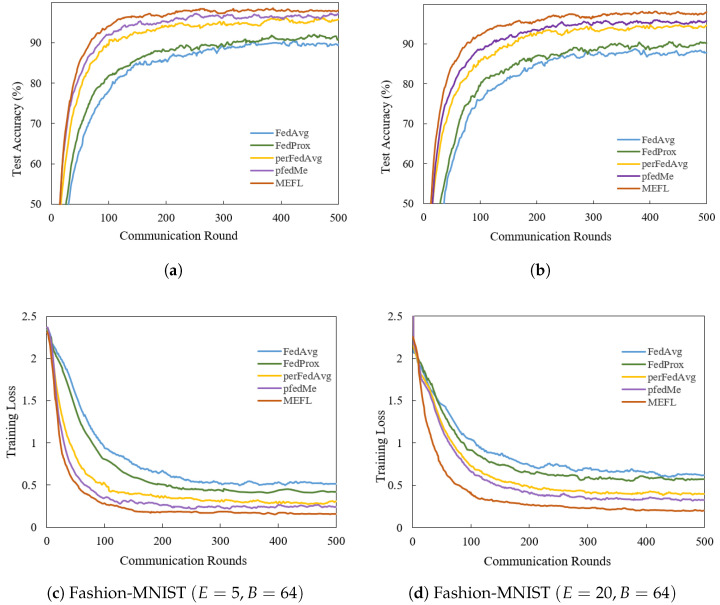
Effect of local epochs on test accuracy and training loss over communication rounds under non-IID Fashion-MNIST. (**a**,**c**) E=5, Fashion-MNIST with imbalanced setting q∼Dir(0.5). (**b**,**d**) E=20, Fashion-MNIST with imbalanced setting q∼Dir(0.5). MEFL outperforms with convergence speed and final accuracy, mitigating the effects of local update skew accumulation.

**Figure 5 entropy-27-00553-f005:**
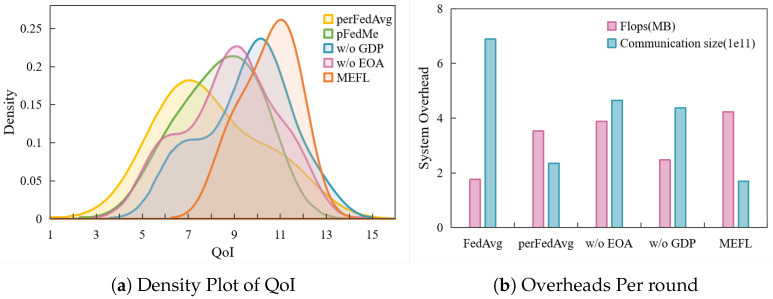
Per-round computational (MFLOPs) and communication ($bytes\times 10^{11}$) overhead for different FL methods on Fashion-MNIST. Density plot of QoI per method on Fashion-MNIST. The peak position of MEFL is further to the right, indicating that the majority of clients have achieved higher personalized gains.

**Figure 6 entropy-27-00553-f006:**
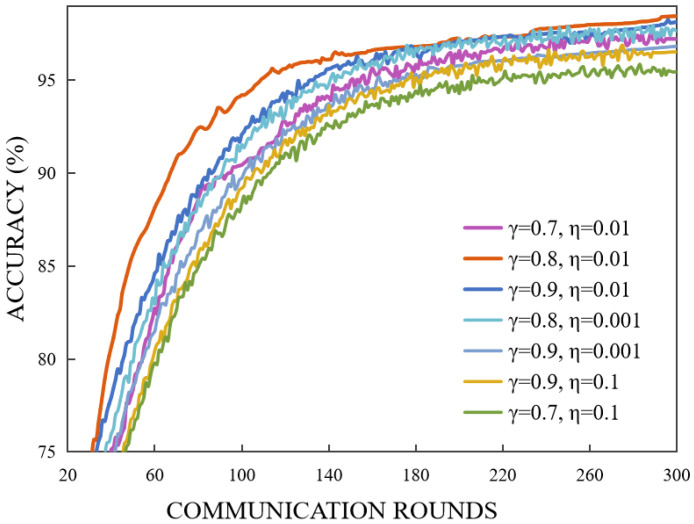
The smoothing factor γ and the global learning rate η determine the smoothing and response speed, thereby influencing the stability and personalization effect of the aggregated weights.

**Table 1 entropy-27-00553-t001:** Symbol definition of latent variables and parameters.

Notations	Description
*i*	Index of client, total number of clients *N*.
*t*	Global communication round index, total number *T*.
τ	Local update step (epochs) index per round, total number *E*.
*P*	Sample fraction of data used as a support set for each client.
$w_{g}^{t}$	Global model parameters at round *t*, $w_{g}^{0}$ at the beginning.
$w_{i}^{t,\tau }$	Local model parameters on client *i* after τ local update steps in round *t*.
$g_{i}^{t}$	Gradient of local model on client in round *T*.
G	The estimated global gradient aggregated from previous rounds.
$S_{i}$	The similarity score of client *i*, with cosine similarity between local model parameter and global gradient vectors.
$\rho_{i}$	The weight of contribution in global aggregation.
(α,β)	Local meta learning rate: inner adaptation rate, meta update rate.
η	Global learning rate.
γ	EMA smoothing factor.
λ	Variance smoothing factor.

**Table 2 entropy-27-00553-t002:** Test accuracy (%) results and standard deviation for different approaches with three support fractions *S*. For IID and non-IID settings, including Fashion-MNIST [45] and CIFAR-10 [46], we run the models until they achieve stable convergence, respectively.

Dataset	P	FedAvg	w/o EOA	w/o GDP	MEFL
	20%	66.51±3.25	72.58±2.36	71.53±2.27	**76.15 ± 1.77**
CIFAR-10	60%	67.29±2.06	72.48±1.87	72.53±1.52	**76.26 ± 1.28**
	90%	68.34±1.67	73.52±1.32	73.64±1.06	**76.58 ± 0.85**
Fashion-	20%	88.72±0.71	94.83±0.62	93.58±0.43	**97.46 ± 0.32**
MNIST	60%	89.31±0.83	95.71±0.71	94.53±0.55	**97.63 ± 0.27**
(Non-IID)	90%	90.16±1.03	96.32±0.45	96.36±0.38	**97.82 ± 0.22**
Fashion-	20%	98.85±0.56	99.28±0.45	99.12±0.26	**99.54 ± 0.17**
MNIST	60%	99.04±0.43	99.35±0.17	99.33±0.19	**99.61 ± 0.03**
(IID)	90%	99.13±0.36	99.34±0.34	99.52±0.22	**99.59 ± 0.05**

Bold fonts indicate better performances.

**Table 3 entropy-27-00553-t003:** The influence of local training epochs *E* and batch size *B* on model stability and divergence with a quantitative comparison table in terms of test accuracy and standard deviation.

Methods	E=5,B=64	E=10,B=64	E=10,B=128	E=20,B=128
FedAvg	88.93 ± 1.25	**89.53 ± 0.83**	89.12 ± 0.94	88.76 ± 1.02
FedProx	90.41 ± 0.93	**91.54 ± 0.76**	90.32 ± 0.81	89.83 ± 0.85
pfedMe	96.45 ± 0.62	**97.83 ± 0.32**	96.83 ± 0.47	96.54 ± 0.34
perFedAvg	94.95 ± 0.78	**96.12 ± 0.42**	95.26 ± 0.48	94.39 ± 0.38
MEFL	97.86 ± 0.32	**98.21 ± 0.15**	97.96 ± 0.21	97.53 ± 0.17

Bold fonts indicate better performances.

**Table 4 entropy-27-00553-t004:** Convergence accuracy and number of communication rounds to reach accuracy milestones for Meta-GDP and EOA separately on Fashion-MNIST (non-IID) (E=10, B=64, P=80%).

Methods	Communication Rounds			Convergence Accuracy
	70%	85%	90%	
FedAvg	65	175	463	90.18% ± 1.02%
w/o GDP	31	58	185	96.37% ± 0.36%
w/o EOA	33	62	196	96.31% ± 0.43%
MEFL	**25**	**47**	**68**	**98.03% ± 0.23%**

Bold fonts indicate better performances.

## Data Availability

We used public CIFAR datasets to evaluate the performance of the proposed method. The CIFAR-10 public datasets can be freely downloaded from the web page at https://www.cs.toronto.edu/~kriz/cifar.html, accessed on 18 April 2025. Fashion-MNIST (Fashion-MNIST) is a canonical computer vision dataset that better aligns with real-world image classification challenges. It can be freely downloaded from the website https://github.com/zalandoresearch/fashion-mnist/tree/master/data/fashion, accessed on 18 April 2025. Sentiment140 consists of Twitter messages with emoticons, and the official link regarding the resources about how it was generated is https://www.kaggle.com/datasets/kazanova/sentiment140, accessed on 18 April 2025.

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
