# Peer review of "MEFL: Meta-Equilibrize Federated Learning for Imbalanced Data in IoT"

_entropy, 2025, doi:10.3390/e27060553_

Round 1
Reviewer 1 Report
Comments and Suggestions for Authors
The paper entitled “MEFL: Meta-Equilibrize Federated Learning for Imbalanced Data in IoT” proposes Meta-Equilibrized Federated Learning (MEFL), which integrates meta-learning with gradient-descent preservation and an equilibrated optimisation aggregation mechanism based on gradient similarity and variance weighted adjustment.
Although the paper is reasonably structured and the topic is interesting, a few issues are identified. Here are some issues and suggestions that need considering to improve the quality of the manuscript:
- Please remove abbreviations from the abstract (e.g., non-IID).
- The introduction clearly defines the importance of this research, however the research question remains vague. Please clarify the research questions in the introduction.
- The related work section 2.2. Mate Learning, first line correct (Iot) to (IoT). Also, the related work section would benefit from a summary of the gap your research is trying to fill.
- The added mechanisms (like EMA, gradient similarity, and GDP tracking) potentially increase the accuracy and reduce communication, but has the computational overhead evaluated?
- Rename ambiguous variables (e.g., avoid wt(τ)_i style notation without clear indexing logic). Also a table of variables would be useful.
- The discussion around the effects of varying local epochs and batch sizes would benefit from quantifiable comparisons or additional statistical insight to support claims about model stability and divergence.
- The fairness and overhead section mentions communication cost savings but lacks computational (time, resources, etc.) overhead that may be introduced by MEFL’s additional components (e.g., GDP and EOA).
- The ablation study is informative, but clearer visualisations (e.g., bar charts or tables with deltas) would help highlight the key finding more intuitively.
- Proofreading is required as there are some typos, grammatical and punctuation errors. Also, it would be appropriate to number the sections. Please check how IoT written, make sure consistency.
Reviewer 2 Report
Comments and Suggestions for Authors
Summary:
The manuscript proposes a new Meta-Equilibrized Federated Learning (MEFL) by integrating meta-learning with gradient-descent preservation and an equilibrated optimization aggregation mechanism that takes into account gradient similarity and variance for adjusting weights in general gradient updates. This helps resolving inconsistency between global and local objectives and enable modifying it to meet the requirements and specifications for local clients.
General concept comments:
- It will benefit if you mention what metrics are used to measure level of local personalization. In the equations, S needs to be explained more.
- It is not how this will reflect similarity.
- It will be interesting to know how the algorithm performs by changing its own hyperparameters (i.e. smoothing factor).
Specific comments:
- Correct Iot to IoT in the page 10.
- It will be great if you add more recent citations in the article.
